# Effect of Intermittent Hypoxia on Metabolic Syndrome and Insulin Resistance in the General Male Population

**DOI:** 10.3390/medicina57070668

**Published:** 2021-06-29

**Authors:** Jung-Yup Lee, Chan-Won Kim, Kyung-Chul Lee, Jae-Hyuk Lee, Sung-Hun Kang, Sung-Won Li, Kyubo Kim, Seok-Jin Hong

**Affiliations:** 1Department of Otolaryngology-Head and Neck Surgery, Kangbuk Samsung Hospital, Sungkyunkwan University School of Medicine, Seoul 03181, Korea; leejaay@gmail.com (J.-Y.L.); fess0101@hanmail.net (K.-C.L.); skyphoenix@hanmail.net (J.-H.L.); 2Center for Cohort Studies, Total Healthcare Center, Kangbuk Samsung Hospital, Sungkyunkwan University School of Medicine, Seoul 03181, Korea; chanwon.kim75@gmail.com; 3Department of Biomedical Sciences, College of Medicine, Hallym University, Chuncheon 24252, Korea; D18024@hallym.ac.kr; 4Department of Otorhinolaryngology-Head and Neck Surgery, Dongtan Sacred Heart Hospital, Hallym University College of Medicine, Hwaseong 18450, Gyeonggi-do, Korea; sw9731@hanmail.net; 5Department of Otorhinolaryngology-Head and Neck Surgery, Kangdong Sacred Heart Hospital, Hallym University College of Medicine, Hwaseong 18450, Gyeonggi-do, Korea

**Keywords:** obstructive sleep apnea, hypoxia, metabolic syndrome, insulin resistance

## Abstract

*Background and objectives:* Obstructive sleep apnea (OSA) is closely associated with insulin resistance (IR) and is an independent risk factor for incident type 2 diabetes mellitus (T2DM). Most studies evaluate the correlation between OSA and IR in only obese or T2DM patients. Therefore, we tried to investigate the effect of OSA on metabolic syndrome and IR in the general healthy male population. *Materials and Methods:* 184 subjects who visited a preventive health examination program were recruited for this study. All subjects received overnight polysomnography by a portable device (Watch-PAT 200). We examined several metabolic parameters and a homeostasis model of assessment for insulin resistance index (HOMA-IR). The subjects were divided into three groups by AHI (Apnea-hyponea index): normal group (AHI < 5), mild OSA group (5 ≤ AHI < 15), and moderate-severe OSA group (AHI ≥ 15). They were also divided into two groups according to minimum oxygen saturation: low group, Min-SpO2 < 88%; and high group, Min-SpO2 ≥ 88%. *Results:* Parameters of metabolic syndrome, including waist circumference, systolic and diastolic blood pressure, triglyceride, and high-density lipoprotein cholesterol showed significant differences among the AHI groups. Furthermore, HOMA-IR showed significant differences among the AHI groups. Those parameters, including metabolic syndrome and HOMA-IR, also showed differences between Min-SpO2 groups. *Conclusions:* In summary, this study helps confirm that AHI is associated with HOMA-IR in the general male population. Furthermore, the severity of AHI correlated with the parameters of metabolic syndrome. Therefore, AHI might be an indicator for evaluating both T2DM and metabolic syndrome, even in the general male population.

## 1. Introduction

Obstructive sleep apnea (OSA) is characterized by repeated partial or complete pharyngeal collapse during sleep leading to intermittent hypoxia (IH), sleep fragmentation, and excessive daytime sleepiness [1]. In addition, IH enhances sympathetic tone leading to increase blood sugar levels by decreasing glucose effectiveness and insulin sensitivity [2]. The decrease in physical activity associated with daytime sleepiness and sleep deprivation can cause alterations in metabolic system such as insulin resistance (IR), and the increased pro-inflammatory states and cytokine levels may also lead to a severe state of IR in patients with OSA [3,4]. In addition, one proposed molecular mechanism is through the α-subunit of hypoxia-inducible factor 1 (HIF-1α), as the oxygen-sensitive protein has been suggested to be involved in the regulation of metabolic processes and development of IR [5]. These disruptions can trigger a series of events related to the activation of the sympathetic system, systemic inflammation, and oxidative stress, all of which can be important in increasing the risk of hypertension, metabolic syndrome, and type 2 diabetes mellitus (T2DM) [6].

Approximately 50% of obese people with metabolic syndrome have OSA [7]. The occurrence of OSA is even higher in obese patients with T2DM and severe obesity [8]. Although previous epidemiological studies suggest that OSA may be an independent risk factor for incident T2DM, there are some conflicting results on the association between OSA and IR, especially based on weight status [9]. Since IR is one of the characters of T2DM and can finally result in the development of T2DM, it is imperative to better understand its relationship with OSA [1]. Most studies evaluate the correlation between OSA and IR in only obese or T2DM patients, and few studies have evaluated this association in nondiabetic healthy people according to central obesity status. Thus, we hypothesized that OSA and IR could affect not only obese or T2DM patients but also non-diabetic healthy people. Accordingly, we aimed to evaluate the relationship between OSA and IR in healthy non-diabetic adults with or without central obesity status.

## 2. Materials and Methods

One hundred eighty-four subjects who visited Kangbuk Samsung Hospital Total Healthcare Screening Center from January 2013 to April 2014 were recruited for this study. These subjects visited the Healthcare Screening Center to receive an overall general health examination. All subjects received an overnight sleep study by portable device for diagnosing OSA. We excluded subjects who had any of the following conditions at baseline: (1) previous history of surgery for upper airway tract; (2) medication on hypertension, T2DM, or any other systemic diseases; (3) previous history of using devices of continuous positive airway pressure; and (4) female subjects. This study was approved by the Institutional Review Board of the Kangbuk Samsung Hospital (No.KBSMC 2014-11-013; 11 December 2014).

### 2.1. Evaluation of Sleep Apnea

Portable device monitoring was performed with Watch-PAT 200 (Itamar Medical, Israel) to evaluate and document sleep apnea parameters, including total sleep time, minimal oxygen saturation (Min-SpO2), apnea-hypopnea index (AHI), respiratory disturbance index (RDI), and oxygen desaturation index (ODI) [10].

The Watch-PAT 200 is a four-channel non-invasive home device. Using a peripheral arterial tonometry (PAT) finger plethysmograph and a standard SpO2 probe, the Watch-Pat 200 records the PAT signal, heart rate, and pulse oximetry, as well as actigraphy from the inbuilt actigraph. PAT signal estimates the arterial pulsatile volume changes of finger that are regulated by α-adrenergic innervation of the smooth muscles of vasculature of finger and thus reflects sympathetic nervous system activity. This augmentation in sympathetic activity accompanies the increase in heart rate and desaturation at the termination of respiratory events. Thus, the Watch-PAT 200 indirectly detects apnea/hypopnea events by identifying surges of sympathetic activation related to the termination of those events. Following the sleep study, the recordings are automatically downloaded and analyzed in an offline procedure using an automatic computerized algorithm. The AHI was defined by the sum of apnea and hypopnea events per each hour of sleep, with the minimal criteria for OSA diagnosis as AHI ≥ 5. Sleep apnea-hypopnea syndrome and its severity were evaluated with international diagnostic criteria: AHI of 5.0–14.9 as mild, 15.0–29.9 as moderate, and ≥30 as severe OSA.

We classified all patients in this study by several parameters for group evaluation. First, patients were classified into three groups by AHI: control group (AHI < 5), mild OSA group (5 ≤ AHI < 15), and moderate-severe OSA group (AHI ≥ 15). Second, they were divided into two groups according to minimum oxygen saturation: low group, Min-SpO2 **<** 88%; and high group, Min-SpO2 **≥** 88%. Finally, they were classified into three groups by ODI: ODI < 5, 5 ≤ ODI < 15, and ODI ≥ 15.

### 2.2. Examination of Metabolic Parameters

Venous blood was drawn following 12 h overnight fast for measurement of plasma fasting blood glucose (FBG), glycosylated hemoglobin (HbA1c), total cholesterol (TC), serum triglyceride (TG), high-density lipoprotein cholesterol (HDL), low-density lipoprotein cholesterol (LDL), true insulin, and proinsulin. IR was estimated with the homeostasis model of assessment for insulin resistance index (HOMA-IR), which can be calculated with the Oxford University online calculator (http://www.dtu.ox.ac.uk/homacalculator/, accessed on 7 June 2015) [11]. HOMA % 𝑆 represents values of 100% in normal adults when using currently available assays for insulin or C-peptide. HOMA-IR is the reciprocal of HOMA % 𝑆. The validity and accuracy of these measurements have been verified, and HOMA-IR was confirmed with a very high correlation of the glucose clamp test.

### 2.3. Anthropometric Measurements

Blood pressure (BP) was checked after each patient waited for 5 min in a sitting position to acquire both systolic BP (SBP) and diastolic BP (DBP). BP check was obtained twice at 3 min interval, and the mean value was used in the data analysis. Body weight and height were obtained to the nearest 0.1 kg and 0.1 cm. Body mass index (BMI) was calculated as weight in kilograms divided by height in meters squared: weight/height^2^ (kg/m^2^). Waist circumference (WC) was checked at the midpoint between the lower rib margin and the iliac crest in the standing position. Obesity was defined as BMI ≥ 25 kg/m^2^ according to the Asian-specific BMI cut offs from the World Health Organization report [12].

### 2.4. Statistical Analysis

The statistics was calculated using Windows SPSS version 24.0 (SPSS Inc. Chicago, IL, USA) in this study, and *p* values below 0.05 were considered statistically significant. Analysis of variance (ANOVA) was performed following logarithmic transformation to compare between group differences. Analysis of covariance (ANCOVA) was also used to compare insulin resistance and components of the metabolic syndrome after adjusting for age. Furthermore, paired *t*-test was used to determine significant differences among groups. The sample normality of our study was tested by using Shapiro–Wilk tests.

## 3. Results

This study included 184 healthy males. The demography of patients in this study is shown in Table 1. There were 27 patients (14.7%) in the normal group, 63 (34.2%) in the mild OSA group, and 94 (51.1%) in the moderate-to-severe OSA group.

The prevalence of metabolic syndrome was compared among different groups. A progressive increase in WC was observed across the three groups. The WC in the normal, mild OSA, and moderate-to-severe OSA groups were 82.78 ± 6.95, 86.69 ± 5.76, and 90.54 ± 7.35, respectively (*p* < 0.05) (Table 2). SBP and DBP in the OSA group were significantly higher than those in the normal group (SBP: 115.12 ± 10.97 in the moderate-to-severe OSA group, 112.67 ± 11.16 in the mild OSA group, and 107.51 ± 7.91 in normal group, *p* < 0.001, DBP: 77.90 ± 9.53 in the moderate-to-severe OSA group, 75.16 ± 8.58 in the mild OSA group, and 69.22 ± 6.46 in the normal group, *p* < 0.05) (Table 2). In addition, there were significant differences in TG and HDL levels among the OSA group (TG: 162.63 ± 88.44 in the moderate-to-severe OSA group, 118.63 ± 54.55 in the mild OSA group, and 106.96 ± 35.40 in the normal group, *p* < 0.001, HDL: 47.77 ± 10.33 in the moderate-to-severe OSA group, 54.14 ± 13.61 in the mild OSA group, and 55.56 ± 11.44 in the normal group, *p* < 0.05). However, no statistical difference was observed in plasma FBG levels (Table 2). A statistically significant correlation of HOMA-IR with AHI was observed (Figure 1) and age-adjusted HOMA-IR in AHI of <5, 5–14, ≥15 were 1.12 (0.90–1.40), 1.30 (1.13–1.49), and 1.83 (1.62–2.05), respectively. (*p* < 0.01). According to OSA and central obesity status defined by WC, the age-adjusted mean value for HOMA-IR was significantly higher among those without central obesity while there were no significant differences in the HOMA-IR among participants with central obesity groups (Table 3). There were 63 patients (34.2%) in the high Min-SpO2 group and 121 patients (65.8%) in the low Min-SpO2 group. In both groups, a statistically significant correlation with WC, HDL, TG, and HOMA-IR was observed. However, there was no differences in SBP, DBP, and FBG (Table 4). Similar to these Min-SpO2 results, HOMA-IR showed significant difference according to ODI (Appendix A).

## 4. Discussion

OSA is a very frequent sleep disorder related to cardiovascular disease and T2DM. Although obesity is a main determinant, and OSA is related to T2DM independent of obesity and other confounding factors such as sex, age, and BMI [9]. We tried to examine the relationship between AHI and parameters of metabolic syndrome in the general population and not in obese or T2DM patients. Results showed that the parameters of metabolic syndrome, including WC, BP, HDL, and TG, were significantly different according to the groups that were divided based on the severity of AHI. Moreover, HOMA-IR was significantly greater in the moderate-to-severe OSA group than in the mild OSA group.

Because OSA is often accompanied by “metabolic syndrome”, the two conditions may causally correlate, and even “Z syndrome” was named for coexisting OSA and metabolic syndrome. Zhang et al. reported that OSA patients with lower SpO2 were more impressionable to IR, which could contribute the coexistence of OSA and metabolic syndrome [13].

The HOMA model is a commonly used method to measure IR. It is a model of the association between glucose and insulin dynamics that anticipates fasting steady-state glucose and insulin concentrations for a wide range of possible combinations of IR and β-cell function [14]. The HOMA model is an easy method for evaluating insulin sensitivity, and it is connected with the results of the glucose clamp test in patients with mild diabetes without hyperglycemia. Multiple studies show a reasonable correlation between HOMA-IR and the euglycemic clamp method that has been the gold standard [15]. Compared with the clamp method, HOMA-IR is simple and time-efficient because it only needs a single blood test assayed for basal state insulin and glucose.

Although mounting studies reported that OSA may be a risk factor for T2DM, previous epidemiological studies showed inconsistent findings regarding the association between OSA and FBG, fasting insulin, or HOMA-IR representing abnormal glucose metabolism. In a study involving 270 middle-aged adults (37–52 years old) without T2DM, Ip et al. found that AHI was a significant independent determinant of fasting insulin and HOMA-IR after controlling for BMI and age, but FBG level did not differ significantly according to the severity of AHI [16]. One previous study of 1344 adults (aged older than 40 years) displayed that FBG was significantly higher in non-obese participants with OSA than in those without OSA, while HOMA-IR and fasting insulin were higher in the obese OSA group than in the obese non-OSA group [17]. In a study involving predominantly non-obese adults, Kritikou et al. found that OSA was significantly associated with HOMA-IR [18]. Likewise with studies of IP et al. and Kritikou et al., our results showed that the relationship between OSA and HOMA-IR differed by weight status with significantly higher HOMA-IR only in the non-obese OSA group, implying that OSA affects insulin sensitivity without preexisting metabolic disturbances of obesity [17]. We supported this insight using quantifying excessive body fat with a variety of measures. Our study results agree with those of Ip et al., who revealed that AHI and Min-SpO2 could significantly determine the fasting insulin and HOMA-IR levels [16]. These inconsistent findings could be due to the difference in study populations and methodology. Instead of targeting only obese subjects or patients suspected of having OSA, our study examined the general population who underwent portable sleep apnea study as part of a preventive health examination program.

Several studies showed possible mechanisms for a causal relationship. Polka et al. reported that IH-induced IR impairs β-cell function, increases hepatocyte glucose output, and increases oxidative stress of the pancreas in C57BL6/J mice [19]. Newer evidence also revealed that chronic IH induced IR in lean mice by promoting the activation of skeletal muscle adenosine monophosphate (AMP)-activated protein kinase [20]. Furthermore, another mechanism proposed that hypoxia in adipose tissue may result in adipocyte cell death, thus causing a plasma free fatty acids increasing, favoring IR [21]. IH also induces an increase in blood TG concentration via reduced adipose tissue lipoprotein lipase activity and a consequent decrease in lipoprotein clearance [22].

From a biochemical molecular standpoint, HIF-1α may play a possible mediating role between OSA and IR. HIF-1α is a subunit of the heterodimeric transcription factor HIF-1 and is a crucial regulator of oxygen metabolism homeostasis. Many studies show substantiating evidence of HIF-1α participating in the inflammatory process caused by the intermittent hypoxia observed in OSA, with HIF-1α levels, significantly increased in severe OSA patients [23,24,25]. Gabryelska et al. even proposed the utilization of HIF-1α as a diagnostic marker of OSA, after excluding other disorders with chronic hypoxia [26].

Moreover, HIF-1α is not only involved in OSA, but it is also linked to IR and diabetes. A C → T non-synonymous single nucleotide polymorphism of HIF-1α gene has been indicated to having a protective effect against diabetes in the Japanese and Hungarian population [27,28]. An animal study showed that a HIF inhibitor may improve IR in high-fat diet mice [29], while in humans, a recent randomized controlled trial discovered that two weeks of eight hours nightly of continuous positive airway pressure increased insulin sensitivity in prediabetes [30]. Considering these results in conjunction with a study where increased serum levels of HIF-1α seen in severe OSA patients were decreased after two months of continuous positive air pressure (CPAP) therapy [25], treating hypoxia may be a putative treatment method for metabolic syndrome via the HIF-1α pathway. However, a pilot study revealed that just one night of CPAP was not sufficient to relieve such pathological changes, and this suggests that long-term CPAP treatment was needed for a protective effect [31].

Estimating hypoxia and the evaluation of OSA rely on polysomnography as ever. Min-SpO2 and AHI are known as crucial factors in the diagnosis of OSA. However, there are some limitations for AHI in investigating the total effect of apnea or hypopnea because AHI only represents the frequency of apneic and hypopnic events and does not reflect the specific duration of each event [32]. In this study, subjects of the low Min-SpO2 group showed significantly higher WC, HDL, TG, and HOMA-IR than those of the high Min-SpO2 group. Therefore, Min-SpO2 might be more clinically important in investigating the degree and results of OSA and metabolic syndrome.

The strength of the current study is that we evaluated a relatively large sample size with a suitable representation of general population, although it was a cross-sectional study. The major limitation of this study was that the results were based on correlation analysis, thus only demonstrating a superficial relationship among OSA severity, IR, and parameters of metabolic syndrome; moreover, it is far away from clarifying the exact relationship with each other. The second limitation of this study was that we included only male subjects. Because the Healthcare Screening Center is visited by mostly male employees who work nearby, nearly 95% of the subjects who visited our center were men. We decided to exclude female subjects for a more homogenous study population. Further studies with both male and female subjects are needed to truly represent the general population.

## 5. Conclusions

In summary, this study helps to confirm that AHI is associated with HOMA-IR in the general population. Furthermore, the severity of AHI correlated with the parameters of metabolic syndrome, including WC, BP, HDL, and TG. Therefore, AHI might be an indicator for evaluating both T2DM and metabolic syndrome, even in the general population. Further large-scale studies detailing the correlation between OSA and metabolic syndrome are warranted.

## Figures and Tables

**Figure 1 medicina-57-00668-f001:**
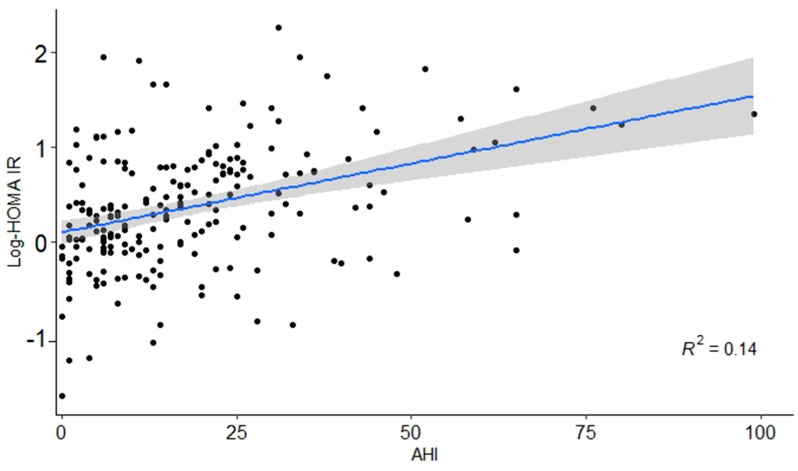
Linear regression between AHI (Apnea-hyponea index) and HOMA-IR (Homeostasis model of assessment for insulin resistance index). AHI and HOMA-IR showed significant correlation.

**Table 1 medicina-57-00668-t001:** Demography of subjects in the study.

	Mean
Age (years)	47.9 ± 8.2
BMI	25.5 ± 3.0
WC (cm)	88.4 ± 7.6
SBP (mmHg)	113.4 ± 11.1
DBP (mmHg)	75.8 ± 9.3
TG (mg/dL)	139.4 ± 74.6 (91.5 to 172)
HDL (mg/dL)	50.8 ± 12.1
FBG (mg/dL)	101.5 ± 17.7
AHI	19.2 ± 16.5 (6.8 to 25)

BMI, body mass index; WC, waist circumference; SBP, systolic blood pressure; DBP, diastolic blood pressure; TG, triglyceride; HDL, high-density lipoprotein cholesterol; FBG, plasma fasting blood glucose; and AHI, apnea-hypopnea index.

**Table 2 medicina-57-00668-t002:** Parameters of metabolic syndrome according to OSA groups.

	AHI < 5 (*n* = 27)	5 ≤ AHI < 15 (*n* = 63)	AHI ≥ 15 (*n* = 94)	*p*-Value
WC *	82.78 ± 6.95	86.69 ± 5.76	90.54 ± 7.35	<0.001
SBP *	107.51 ± 7.91	112.67 ± 11.16	115.12 ± 10.97	0.005
DBP *	69.22 ± 6.46	75.16 ± 8.58	77.90 ± 9.53	<0.001
HDL *	55.56 ± 11.44	54.14 ± 13.61	47.77 ± 10.33	<0.001
TG *	106.96 ± 35.40	118.63 ± 54.55	162.63 ± 88.44	<0.001
FBG	99.00 ± 12.59	97.17 ± 9.41	101.40 ± 13.98	0.137

OSA, obstructive sleep apnea; WC, waist circumference; SBP, systolic blood pressure; DBP, diastolic blood pressure; TG, triglyceride; HDL, high-density lipoprotein cholesterol; FBG, plasma fasting blood glucose; and AHI, apnea-hypopnea index. * indicates a *p* value < 0.05.

**Table 3 medicina-57-00668-t003:** HOMA-IR (95% confidence interval) according to OSA groups.

HOMA-IR	AHI < 5 (*n* = 27)	5 ≤ AHI < 15 (*n* = 63)	AHI ≥ 15 (*n* = 94)	*p*-Value
Overall	1.12 (0.90–1.40)	1.30 (1.13–1.49)	1.83 (1.62–2.05)	<0.001
WC < 90	1.09 (0.89–1.34)	1.12 (0.97–1.30)	1.45 (1.25–1.68)	0.023
WC ≥ 90	1.47 (0.80–2.70)	1.80 (1.38–2.36)	2.21 (1.86–2.61)	0.255

Age adjusted means were calculated using analysis of covariance. HOMA-IR, homeostasis model of assessment for insulin resistance index; OSA, obstructive sleep apnea; AHI, apnea-hypopnea index and WC, waist circumference.

**Table 4 medicina-57-00668-t004:** Parameters of metabolic syndrome and HOMA-IR according to minimum oxygen saturation groups.

	Min-Saturation ≥ 88% (*n* = 63)	Min-Saturation < 88% (*n* = 121)	*p*-Value
WC *	85.19 ± 7.71	88.78 ± 8.00	0.002
SBP	112.12 ± 16.42	113.25 ± 8.87	0.511
DBP	74.30 ± 9.96	76.06 ± 10.33	0.224
HDL *	54.81 ± 13.17	50.77 ± 13.17	0.029
TG *	113.74 ± 82.20	144.22 ± 82.20	0.009
FBG	97.99 ± 13.46	100.62 ± 13.96	0.179
HOMA-IR *	1.38 ± 1.34	1.93 ± 1.34	0.002

WC, waist circumference; SBP, systolic blood pressure; DBP, diastolic blood pressure; TG, triglyceride; HDL, high-density lipoprotein cholesterol; FBG, plasma fasting blood glucose; and HOMA-IR, homeostasis model of assessment for insulin resistance index. * indicates a *p* value < 0.05.

## Data Availability

The data used to support the findings of this study are available from the corresponding author upon request.

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
