# Peer review of "Effect of Intermittent Hypoxia on Metabolic Syndrome and Insulin Resistance in the General Male Population"

_medicina, 2021, doi:10.3390/medicina57070668_

Round 1

Reviewer 1 Report

 The authors took up a very interesting topic of the effect of  Effect of intermittent hypoxia on metabolic syndrome and insulin resistance in the general male population. My main concern is very poor bibliography - the authors should perform better literature search. Moreover, I have some suggestions which may increase the quality of the proposed paper:

- Introduction: During the literature search pertaining to obstructive sleep apnea, insulin resistance, and type 2 diabetes mellitus. I have found an appropriate minireview, which discussed these dependencies (10.3389/fphys.2020.01035). The paper may benefit from pertaining to this manuscript in the introduction!
- Introduction: The hypothesis of the study should be added!
- Materials and methods:
a)  This study was approved by the Institutional Review Board of the Kangbuk Samsung Hospital - please provide here the number of the approval

b) what with sample size calculation? 
c) Have you check the normality of obtained data? It should be stated.
d) all p values should be stated with the same accuracy (0.xyz)
- Discussion: In my opinion, the results should be discussed with the results discussed in the aforementioned minireview, especially: a) 10.3390/jcm9051599.
b) 10.1371/journal.pone.0070559
c) 10.20452/pamw.15104
d) 10.17305/bjbms.2016.1579
e) 10.1111/jsr.12995
f) 10.1186/1471-2350-10-79
g) 10.5664/jcsm.8682

Author Response

Title Effect of intermittent hypoxia on metabolic syndrome and insulin resistance in the general male population We would like to thank the reviewers for their constructive criticism, which has helped us improve the manuscript. We have attempted to revise our study more carefully and thoroughly to address all the reviewers’ concerns. The following is a point-by-point response to the reviewers’ comments. We hope the current version of the manuscript is suitable for publication in Medicina. The authors took up a very interesting topic of the effect of Effect of intermittent hypoxia on metabolic syndrome and insulin resistance in the general male population. My main concern is very poor bibliography - the authors should perform better literature search. Moreover, I have some suggestions which may increase the quality of the proposed paper: Q1 - Introduction: During the literature search pertaining to obstructive sleep apnea, insulin resistance, and type 2 diabetes mellitus. I have found an appropriate minireview, which discussed these dependencies (10.3389/fphys.2020.01035). The paper may benefit from pertaining to this manuscript in the introduction! A: Thank you for your valuable suggestion. In accordance with your comment, we have added the minireview to the Introduction, as recommended by the reviewer. (Page 1~2) In addition, one proposed molecular mechanism is through the α-subunit of hypox-ia-inducible factor 1 (HIF-1α), as the oxygen-sensitive protein has been suggested to be involved in the regulation of metabolic processes and development of IR.[5] Q2 - Introduction: The hypothesis of the study should be added! A: We have the following hypothesis to the Introduction. (Page 2) “Thus, we hypothesized that OSA and IR could affect not only obese or T2DM patients but also non-diabetic healthy people.” Q3- Materials and methods: a) This study was approved by the Institutional Review Board of the Kangbuk Samsung Hospital - please provide here the number of the approval A: We have added the IRB number to the manuscript. (Page 2) “This study was approved by the Institutional Review Board of the Kangbuk Samsung Hospital (No.KBSMC 2014-11-013; 11 Dec, 2014 ).” Q4 b) what with sample size calculation? A : Thank you for pointing this out. The sample size was calculated in R 3.2.0 (command: pwr.t.test) The values were effect size=1.1, power=0.9, and level of significance of p=0.05, and gave a sample size of 36 participants. Q5 c) Have you check the normality of obtained data? It should be stated. A: The sample normality of our study was tested by using Shapiro Wilk tests. (Page 3) The sample normality of our study was tested by using Shapiro Wilk tests. Q6 d) all p values should be stated with the same accuracy (0.xyz) A : We agree and revised all the p values with the same accuracy. Q7 - Discussion: In my opinion, the results should be discussed with the results discussed in the aforementioned minireview, especially: a) 10.3390/jcm9051599. b) 10.1371/journal.pone.0070559 c) 10.20452/pamw.15104 d) 10.17305/bjbms.2016.1579 e) 10.1111/jsr.12995 f) 10.1186/1471-2350-10-79 g) 10.5664/jcsm.8682 A : Thank you for your insightful comments. Per your recommendation, we have added the recommended literature to the Discussion. (Page 6~7) From a biochemical molecular standpoint, HIF-1α may play a possible mediating role between OSA and IR. HIF-1α is a subunit of the heterodimeric transcription factor HIF-1 and is a crucial regulator of oxygen metabolism homeostasis. Many studies show substantiating evidence of HIF-1α participating in the inflammatory process caused by the intermittent hypoxia observed in OSA, with HIF-1α levels significantly increased in severe OSA patients [23-25]. Gabryelska et al. even proposed the utilization of HIF-1α as a diagnostic marker of OSA, after excluding other disorders with chronic hypoxia [26]. Moreover, HIF-1α is not only involved in OSA, but it is also linked to IR and diabetes. A C→T non-synonymous single nucleotide polymorphism of HIF-1α gene has been indicated to having a protective effect against diabetes in the Japanese and Hungarian population [27, 28]. An animal study showed that a HIF inhibitor may improve IR in high-fat diet mice, [29] while in humans, a recent randomized controlled trial discovered that two weeks of eight hours nightly of continuous positive airway pressure increased insulin sensitivity in prediabetes.[30] Considering these results in conjuction with a study where increased serum levels of HIF-1α seen in severe OSA patients were decreased after two months of continuous positive air pressure (CPAP) therapy [25], treating hypoxia may be a putative treatment method for metabolic syndrome via the HIF-1α pathway. However, a pilot study revealed that just one night of CPAP was not sufficient to relieve such pathological changes and this suggests that long-term CPAP treatment was needed for a protective effect [31].

Reviewer 2 Report

Good work about metabolic consequences of Obstructive Sleep Apnea. My only comment to the authors is that probably time under 90% of saturation (CT90) could be better indicator of intermittent hypoxia than minimal saturation <94%.

Author Response

Title Effect of intermittent hypoxia on metabolic syndrome and insulin resistance in the general male population We would like to thank the reviewers for their constructive criticism, which has helped us improve the manuscript. We have attempted to revise our study more carefully and thoroughly to address all the reviewers’ concerns. The following is a point-by-point response to the reviewers’ comments. We hope the current version of the manuscript is suitable for publication in Medicina. Good work about metabolic consequences of Obstructive Sleep Apnea. My only comment to the authors is that probably time under 90% of saturation (CT90) could be better indicator of intermittent hypoxia than minimal saturation

Round 2

Reviewer 1 Report

Authors adressed all my coments suficently in present version of the manuscript.

Author Response

Title

Effect of intermittent hypoxia on metabolic syndrome and insulin resistance in the general male population

We would like to thank the reviewers for their constructive criticism, which has helped us improve the manuscript. We have attempted to revise our study more carefully and thoroughly to address all the reviewers’ concerns. The following is a point-by-point response to the reviewers’ comments. We hope the current version of the manuscript is suitable for publication in Medicina.

Minor revision

<Reviewer >

 In this interesting study the authors aimed to investigate the effect of OSA on metabolic syndrome and IR in the general healthy male population. However, I would like to make some comments:

Q1. - In the abstract the authors write ‘They were also divided into two groups according to minimum oxygen saturation: low group, Min-SpO2≤ 84%, and high group, Min-SpO2 > 84%’; however in lines 98- 99 they write ‘ in addition, they were divided into two groups according to minimum oxygen saturation: low group, Min-SpO2< 90%; high group, Min-SpO2 ≥90%’. Which one is correct and why didn’t they select for cut off of Sat 88%?

A: Thank you for pointing this out. It was absolutely our mistake, and Min-SpO2 standard was 90%. We corrected the abstract as you mentioned above.

(Page 1) They were also divided into two groups according to minimum oxygen saturation: low group, Min-SpO2 < 90%, and high group, Min-SpO2 ≥ 90%.

Q2. - Did BMI differ between OSA severity groups? Could BMI affect the results?

A: Although BMI differed by OSA severity, the results did not change materially after adjustment for BMI.

(Page 2) “Thus, we hypothesized that OSA and IR could affect not only obese or T2DM patients but also non-diabetic healthy people.”

Q3.- Do the authors have any data on the association of ODI with different parameters of metabolic syndrome?

A: Thank you for your insightful comments. We analyzed the association between ODI and parameters of metabolic syndrome. We have added the table for supplement.

(Supplementary table )

Supplementary Table 1. Parameters of metabolic syndrome according to ODI groups

ODI < 5

(n=66)

5 ≤ ODI < 15

(n=57)

ODI ≥ 15

(n=61)

p-value

WC*

85.16 ± 6.90

87.93 ± 6.89

91.39 ± 6.90

<0.001

SBP

110.78 ± 10.89

113.68 ± 10.84

115.25 ± 10.92

 0.068

DBP*

72.85 ± 9.06

76.54 ± 9.02

77.97 ± 9.08

 0.005

HDL*

54.41 ± 11.82

50.18 ± 11.77

48.35 ± 11.85

 0.014

TG*

115.68 ± 69.79

137.15 ± 74.76

167.16 ± 72.78

<0.001

FBG*

99.27 ± 12.31

96.41 ± 12.26

102.95 ± 12.34

 0.017

WC, waist circumference; SBP, systolic blood pressure; DBP, diastolic blood pressure;

TG, triglyceride; HDL, high-density lipoprotein cholesterol; FBG, plasma fasting blood glucose;

ODI, oxygen-desaturation index. * indicates a p value < 0.05.
